# Personal and perceived stigma towards mental disorders among attendants of patients with mental illness in selected health facilities of Bangladesh

Shagoofa Rakhshanda[1,2], Labida Islam[1*], Koustuv Dalal[3], Aklima Anwar Mitu[1], Farah Naz Rahman[1,4], Minhazul Abedin[1,5], Abrar Wahab[1], Cinderella Akbar Mayaboti[1], Salim M. Chowdhury[1], AKM Fazlur Rahman[1], Evan Atlantis[6,7,8], Saidur Rahman Mashreky[1,9]

1 Centre for Injury Prevention and Research, Bangladesh (CIPRB), Dhaka, Bangladesh, 2 School of Population Health, UNSW Sydney, Sydney New South Wales, Australia, 3 School of Health Sciences, Mid Sweden University, Sundsvall, Sweden, 4 School of Public Health and Preventive Medicine, Monash University, Melbourne, Australia. 5 Department of Population Health Science, John D. Bower School of Population Health, University of Mississippi Medical Center, Jackson, Mississippi, United States of America, 6 School of Health Sciences, Western Sydney University, Penrith, New South Wales, Australia, 7 Translational Health Research Institute, Western Sydney University, Penrith, New South Wales, Australia, 8 Faculty of Medicine and Health, Discipline of Medicine, Nepean Clinical School, The University of Sydney, Sydney, New South Wales, Australia, 9 Department of Public Health, North South University, Dhaka, Bangladesh

☯ These authors contributed equally to this work.
* labidaislam1721@gmail.com

## Abstract

The objective of this study was to explore the stigma among attendants (close relatives or other caregivers) who brought patients with mental illness to the selected mental health facilities in Bangladesh. This mixed-method study was part of a nationwide survey where respondents were conveniently selected. Cross-sectional data were collected from 176 attendants of patients with mental illness, and qualitative data were collected from 40 respondents. Quantitative data were collected in handheld tablets using the Day's Mental Illness Stigma Scale questionnaire. Qualitative data was collected in recorders, using a semi-structured guideline. The attendants had more stigma around the patients' ability to maintain relationships, hygiene and discretion regarding their illness, with the highest level of stigma towards hygiene issues (score: 5.4). There was more positive attitude towards the treatability, curability, and recovery of the patients due to faith in the professional expertise of the service providers at mental health facilities, with the least level of stigma towards professional expertise (score: 6.4). Older people, females, and educated attendants were generally less stigmatized towards patients with mental illness than their counterparts. This study found that more stigma was present around the domains' relationships, hygiene, anxiety, and visibility, while less stigma was present around the domains' treatability, curability, and recovery of the patients. Further studies can be

**Data availability statement:** All relevant data are within the manuscript and its Supporting Information files.

**Funding:** The study was funded by the Directorate General of Health Services (DGHS) Bangladesh (invitation ref no: DGHS/LD/NCDC/Procurement plan/RPA (GOB) Service/2018-19/ 2018/5217/SP-05) under the name of Saidur Rahman Mashreky as Principal Investigator. The funders had no role in study design, data collection and analysis, decision to publish, or preparation of the manuscript.

**Competing interests:** The authors have declared that no competing interests exist.

conducted to understand the effect of various factors on stigma, to aid in the development of interventions and counselling frameworks.

## Introduction

Globally, mental disorders are leading causes of poor health and disability [1]. According to the World Health Organization (WHO), more than 970 million people worldwide were affected by mental disorders in 2019 [2]. Evidence suggests that this burden accounted for 14.6% of years lived with disability and the loss of 4.9% of Disability-Adjusted Life-Years (DALYs) globally [3]. Over the past decade, several demographic factors played a role in the 13% rise in the global prevalence of mental disorders [4]. The disorder affects females more than males, and about 20% of the children and adolescents worldwide [5,6]. Over 80% of people suffering from mental disorders live in Low and Middle-Income Countries (LMICs), and mental illness accounted for nine percent of the total burden of disease in LMICs in 2017 [7]. While a recent study reported the prevalence of common mental disorders in South East Asia to be as high as 14% [8], there is an absence of any such surveillance in Bangladesh. However, a systematic review of 32 research articles suggested that the burden of common mental disorders varied from about seven percent to 31% among adults and three percent to 23% among children [9]. As Bangladesh has a population of 162 million people and severely lacks healthcare resources, the standard of mental health care remains grossly inadequate [10]. Despite this, identifying barriers to accessing mental healthcare in Bangladesh could help reduce the nationwide health and economic burden of common mental disorders.

Stigma is often referred to as attitudes, behaviours, and beliefs of a person that lead to rejection, avoidance, or fear of those who are different [11]. According to Goffman, stigma is considered to be a complex social process of labeling, devaluing, and discriminating against others [12]. Stigmatization may be instigated by family members, friends, and colleagues, or powerful social groups. The outcome of stigma may be isolation and/or loss of unit cohesion that hamper intimate relationships, leading to further deterioration of mental health. This has a profound effect on the conceptualization of health needs and healthcare-seeking behavior of people with mental disorders [13,14]. Three main types of stigmas have been proposed for mental illness, including self-stigma, personal stigma, and perceived public stigma. Self-stigma is attitudes, perceptions, and behavior of a person with a mental illness towards other people who also suffer from a mental disorder [15].

Stigma of any type towards mental disorders has been found to have a strong negative association with healthcare-seeking behavior [11]. The results of a recently published systematic review suggest that only 16% of patients with a mental disorder in Bangladesh directly seek healthcare services. This delay was likely due to the lack of awareness and perceived public stigma of mental illness, whereby people with mental disorders may have been more concerned about the reactions of others within their community [9]. Studies have often shown that patients with mental illness have

limited cognitive capacity when making decisions regarding seeking healthcare services on their own. During these cases, close relatives or other caregivers ('attendants') would likely assume responsibility for these patients' mental health-care [16]. Thus, stigma among attendants of patients with mental disorders may present another barrier to seeking and accessing mental healthcare and services [16]. In this regard, the present study aims to explore the perception towards mental disorders among the attendants who brought patients with mental illness to the selected mental health facilities in Bangladesh.

## Materials and methods

### Study design and setting

This was a mixed-method study that incorporated both qualitative and quantitative components to address the study objectives. It was part of a nationwide survey to assess mental health-related service availability and readiness in Bangladesh. The study duration was six months from January to June 2019.

Data was collected from 2nd March 2019–20th April 2019, so during this time frame we found 176 attendants. Thus, we considered those 176 attendants as the study sample size. Ten health facilities in Bangladesh that provide inpatient and outpatient services for mental health patients were purposively selected for this study. Out of these ten health facilities, two were selected as specialized mental health care centres in Bangladesh, while the rest of eight medical college hospitals were selected based on the availability of mental health services. The respondents for this study were conveniently selected considering our socio-cultural environment, as mental health conditions are still considered taboo. As such, the attendants of patients who were willing to give consent to take part in this study at the time of interview were selected as the respondents for this study.

### Data collection instrument and procedure

The quantitative data was collected in hand-held Android tablets using REDCap software. Quantitative data was collected from 176 attendants, where 15–20 (as per availability on the day of the interview) respondents were interviewed per facility. Of patients who received inpatient (patients who are admitted in a health facility) and/or outpatient (patients who receive services, without being admitted in any facility) services from the selected tertiary and specialized health facilities from 2nd March 2019–20th April 2019. This Likert Scale was developed by Emer Day (2007), as part of her doctoral dissertation at her university, to identify and measure the relation between mental illness and stigma related to mental illness [12]. The scale was validated in the context of the local setting of Bangladesh by translation and back translation [17,18]. Prior to data collection, a pretest of the questionnaire was conducted in the above-selected health facilities among 30 attendants. The scale was utilized in the study since there aren't any culturally validated measures that measure the stigma associated with mental illness in Bangladesh. The questionnaire sought information regarding the socio-demographic characteristics of the respondents as well as their attitude towards mental illness. However, the main tool that was used in this study was the Day's Mental Illness Stigma Scale questionnaire to assess the level of stigma of the patients' attendants towards mental illness in others. The Scale addressed seven factors (treatability, relation, hygiene, anxiety, visibility, recovery, and professional efficacy) that indicate the various stigmatized behaviors towards a person with mental illness, considering the social behavior of people towards others [12].

To explore the presence of barriers and challenges in seeking services in Bangladesh for mental disorders, qualitative data were collected from 40 respondents from 13th April to 20th April, 2019. As we reached the thematic saturation point after 40 in-depth interviews, we stopped collecting qualitative data. With the judgement of lead researcher, the qualitative study team decided to stop data collection after 40 interviews, as there were no longer generating new themes, insights or any codes relevant to our research question and the redundancy was consistent. This thematic saturation aligns with recommendations from Guest et al. [19] that data saturation can often be achieved within 12–15 interviews for relative

similar participant groups. The qualitative data was collected in recorders, using a semi-structured guideline which was developed by an anthropologist.

Four anthropologists were trained on the data collection tools (both qualitative and quantitative) and the administration of the questionnaire and guidelines. The trainings were provided by the Research Associates and anthropologists who had adequate knowledge of the data collection tools.

The four data collectors were divided into two teams (each consisting of two data collectors). Each team visited five health facilities to collect data. Both quantitative and qualitative data were collected at their convenience of the respondents, at a time when they were available to respond. During the in-depth interview, one data collector led the conversation while another took relevant notes. All in-depth interviews were audio-recorded. The quantitative data gathering process took 25 minutes, whereas the qualitative data collection process took 40 minutes to complete.

## Data analysis

The quantitative data were entered in a predesigned Microsoft Office Excel format, which was later imported into statistical software SPSS version 25. The raw data were checked for completeness and consistency, and then cleaned. Relevant descriptive statistical analysis was also performed. Afterwards, the results were presented in tables and illustrations. The Day's Mental Illness Stigma Scale is a Likert scale where there are a few questions under each of the seven factors/domains. The responses are scored, where a score of one is given to complete disagreement and a score of seven is given to complete agreement. The average of these scores for individual domains was calculated during the analysis of the dataset. Depending on the direction of the questions under each domain, some of the domains (relation, hygiene, anxiety and visibility) were scored positively; a mean score higher than the neutral score of four suggested a negative attitude towards mental illness, implying the presence of stigmatization and some (treatability, recovery and professional efficiency) were scored reversely; a mean score higher than the neutral score suggested a positive attitude towards mental illness and implying the presence of less stigmatization. A mean score of four implies a neutral response to the domain of stigma. The distribution of the scores from Day's stigma scale according to the sociodemographic characteristics was calculated and displayed in a table. Relevant statistical analysis was performed to check the presence of any significant difference among the scores according to the sociodemographic characteristics.

The in-depth interviews were transcribed verbatim and translated into English. The data was then summarized and analyzed manually by two researchers as per a guideline consisting of a list of a priori codes and themes, which were based on literature reviews. Codes were developed starting with a broad and descriptive familiarization. Then they were refined into meaningful categories and then groups into themes. After that, a documentation was made with a codebook. To enhance reliability, two researchers independently coded four of the same initial transcripts, and the results were compared for the establishment of the codes. Any discrepancies or differences in opinions were discussed and resolved by the researchers and inter-coder agreement was assessed periodically by the joint coding an additional transcript every 5 interviews (k = o.82). Thematic analysis was carried out in this study followed Braun and Clarkes' six phase framework for thematic analysis [20], which coincides with the quantitative results of this study. Relevant quotations were used to support the descriptive analysis of the data.

## Ethical consideration

The ethical clearance for this study was obtained from the Ethical Review Board of the Centre for Injury Prevention and Research, Bangladesh (ERC number: CIPRB/ERC/2019/004). Written informed consent was obtained from each study participant. Participants had the right to withdraw from the study at any point during data collection. Attempts were made by the researchers to maintain scientific rigour, accuracy, impartiality, emotional safeguarding, and anonymity where appropriate during data collection. Qualitative data was collected in a closed room to ensure the anonymity of the attendees.

## Result

From the selected ten health facilities, quantitative data were collected from 176 respondents. Among these respondents, 55.1% were males and 44.9% were females (Table 1). Among females, most of them were housewives (36.9%) aged from 31 to 50 years (55.7%) and education up to and above secondary level (67.6%).

Qualitative data were collected from 40 respondents, but among these respondents, there were 12 attendants of outpatients and 8 attendants of inpatients (Table 1). The remaining 20 respondents were either doctors or nurses; thus, they were not included in the sociodemographic characteristics of the respondents of patient respondents. Most of the interviewed respondents were males (80.0%) of less than 30 years (55.0%) with almost evenly distributed proportion of education attainment (45.0% up to primary level education), were self-employed (35.0%), and 40% of the respondents were either brother or sister of the patients.

### Positively scored domains

The mean scores and the standard deviation of the positively scored domains found were: relation (4.7), hygiene (5.4), anxiety (4.7), and visibility (5.1), as shown in Fig 1.

A) Relationship.

About 55% of the respondents agreed ('partial' or higher) that it is difficult to maintain a normal relationship with someone who has a mental illness, and that such relationships are emotionally exhausting, since they are very demanding and not trustworthy (Table 2).

**Table 1. Sociodemographic characteristics of the respondents for quantitative and qualitative data collection.**

| Variable | Category | Type of data collection [n (%)] | |
| --- | --- | --- | --- |
| | | Quantitative [N = 176] | Qualitative [N = 20] |
| Age | ≤ 30 | 67 (38.1) | 11 (55.0) |
| | 31-50 | 98 (55.7) | 9 (45.0) |
| | ≥ 51 | 11 (6.3) | 0 |
| Sex | Male | 97 (55.1) | 16 (80.0) |
| | Female | 79 (44.9) | 4 (20.0) |
| Education | No schooling | 20 (11.4) | 0 (0.0) |
| | Primary | 37 (21.0) | 9 (45.0) |
| | Secondary and above | 119 (67.6) | 11 (55.0) |
| Occupation | Farmer | 14 (8.0) | 1 (5.0) |
| | Self employed | 46 (26.1) | 7 (35.0) |
| | Service holder | 28 (15.9) | 4 (20.0) |
| | Housewife | 68 (38.6) | 4 (20.0) |
| | Unemployed | 16 (9.1) | 3 (15.0) |
| | Others | 4 (2.3) | 1 (5.0) |
| Relationship with the patient | Sibling | | 8 (40.0) |
| | Husband/Wife | | 1 (5.0) |
| | Father/Mother | | 3 (15.0) |
| | Son/Daughter | | 2 (10.0) |
| | Others (Nephew, grandson, sister-in-law) | | 6 (30.0) |

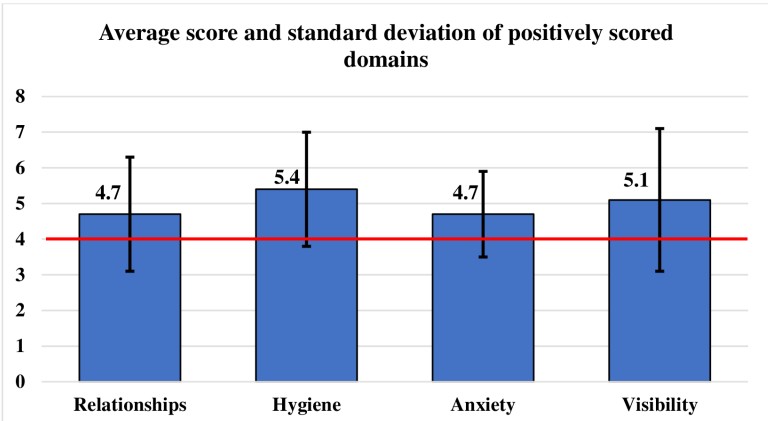

**Fig 1. Average score and standard deviation of positively scored stigma domains among the attendants of inpatients and outpatients, as per Day's Mental Illness Stigma Scale (n = 176).**

In-depth interviews with attendants show that there are several layers to maintaining a relationship with patients with mental illness. Attendants (such as parents, siblings, and spouses) of the patients mentioned that they treat the patients nicely and with compassion, but sometimes it feels like a burden or a responsibility.

According to the respondents from qualitative exploration, the society treated people with mental illness very differently and deleteriously. They further mentioned that due to this fear of damaging a relationship, many do not want to visit the doctor and hide the truth of mental illness within the family.

> *"My daughter-in-law has left my son as she came to know that her husband is suffering from mental illness. She thinks that her husband is going to be mad soon, and there is no cure for it."*

B) Hygiene.

More than 70% of the respondents agreed that people with mental illness do not maintain personal hygiene and tend to ignore their appearance, as shown in Table 2.

In-depth interviews with attendants, doctors as well and nurses suggest that mental patients cannot think rationally, and they are not very conscious about their hygiene issues.

> *"Healthy and normal people are mature and intelligent. But mentally ill persons are not able to groom themselves; that is why they need constant care and support from other members of the family. However, this task is very tiresome, time-consuming, and most of the time very repulsive."*

C) Anxiety.

More than 65% of the respondents agreed that they feel anxious, uncomfortable, and nervous around people with mental illness.

Through qualitative data, it showed that most of the attendants mentioned that they do not feel comfortable or relaxed around the person with a mental illness because they think that they will cause them harm. One doctor narrated that,

> *"My patient sometimes uses slang with me. He even beats me and breaks things. But all these have decreased now. It was unbearable in the past, but it is more tolerable now."*

**Table 2. Distribution of scores according to each of the positively scored domains.**

| Domain | Statements | Agreement level | | | | | | |
|---|---|---|---|---|---|---|---|---|
| | | Complete disagree n (%) | Disagree n (%) | Partial disagree n (%) | Neutral n (%) | Partial agree n (%) | Agree n (%) | Complete agree n (%) |
| Relationship | Impossible to have a normal relationship with people with a mental illness | 42 (23.9) | 16 (9.1) | 13 (7.4) | 1 (0.6) | 47 (26.7) | 41 (23.3) | 16 (9.1) |
| | Finds difficulty in trusting people with a mental illness | 24 (13.6) | 12 (6.8) | 10 (5.7) | 5 (2.8) | 45 (25.6) | 40 (22.7) | 40 (22.7) |
| | A close relationship with someone with a mental illness is likely to be living on an emotional roller coaster | 34 (19.3) | 6 (3.4) | 14 (8.0) | 6 (3.4) | 39 (22.2) | 44 (25.0) | 33 (18.8) |
| | A personal relationship with someone with a mental illness would be too demanding | 23 (12.1) | 8 (4.5) | 19 (10.8) | 2 (1.1) | 27 (15.3) | 55 (31.3) | 42 (23.9) |
| | Mental illnesses prevent people from having normal relationships with others | 8 (4.5) | 7 (4.0) | 14 (8.0) | 10 (5.7) | 36 (20.5) | 51 (29.0) | 50 (28.4) |
| Hygiene | Negligence to own appearance | 22 (125) | 12 (6.8) | 10 (5.7) | 1 (0.6) | 17 (9.7) | 52 (29.5) | 62 (35.2) |
| | Ignorance of self-hygiene, such as bathing and using deodorant | 17 (9.7) | 12 (6.8) | 10 (5.7) | 3 (1.7) | 21 (11.9) | 38 (21.6) | 75 (42.6) |
| | Not grooming themselves properly | 26 (14.8) | 8 (4.5) | 10 (5.7) | 3 (1.7) | 18 (10.2) | 59 (33.5) | 52 (29.5) |
| | Need to take better care of grooming (bathe, clean teeth, use deodorant) | 2 (1.1) | 8 (4.5) | 10 (5.7) | 2 (1.1) | 12 (6.8) | 33 (18.8) | 109 (61.9) |
| Anxiety | Anxiety and discomfort while around someone with a mental illness | 23 (13.1) | 19 (10.8) | 16 (9.1) | 1 (0.6) | 45 (25.6) | 36 (20.5) | 36 (20.5) |
| | Anxiety and nervousness when around someone with a mental illness | 12 (6.8) | 9 (5.1) | 19 (10.8) | 7 (4.0) | 48 (27.3) | 44 (25.0) | 37 (21.0) |
| | Worried for ended up telling upsetting things to someone with a mental illness while talking | 10 (5.7) | 5 (2.8) | 10 (5.7) | 3 (1.7) | 38 (21.6) | 59 (33.5) | 51 (29.0) |
| | Do not feel relaxed when around someone with a mental illness | 13 (7.4) | 21 (11.9) | 24 (13.6) | 2 (1.1) | 37 (21.0) | 35 (19.9) | 44 (25.0) |
| | Worried for their physical harm while around someone with a mental illness | 8 (4.5) | 9 (5.1) | 8 (4.5) | 5 (2.8) | 42 (23.9) | 49 (27.8) | 55 (31.3) |
| | Feeling unsure about what to say or do while around someone with a mental illness | 38 (21.6) | 30 (17.0) | 34 (19.3) | 6 (3.4) | 19 (10.8) | 35 (19.9) | 14 (8.0) |
| | Feeling nervous and uneasy when near someone with a mental illness] | 24 (13.6) | 16 (9.1) | 18 (10.2) | 7 (4.0) | 40 (22.7) | 41 (23.3) | 30 (17.0) |
| Visibility | Easy to recognize the symptoms of mental illnesses | 35 (19.9) | 26 (14.8) | 25 (14.2) | 11 (6.3) | 35 (19.9) | 30 (17.0) | 14 (8.0) |
| | Cannot identify a mental illness without telling | 13 (7.4) | 16 (9.1) | 25 (14.2) | 4 (2.3) | 37 (21.0) | 49 (27.8) | 32 (18.2) |
| | Identify mental illness by the way he or she acts | 25 (14.2) | 19 (10.8) | 18 (10.2) | 13 (7.4) | 28 (15.9) | 54 (30.7) | 19 (10.8) |
| | Identify someone with a mental illness by the way he or she talks | 32 (18.2) | 20 (11.4) | 22 (12.5) | 3 (1.7) | 36 (20.5) | 50 (28.4) | 13 (7.4) |

One of the respondents stated that,

*"Mentally ill persons are evaluated separately than the normal persons of the society. Their family members always stay in a fear with them. Because mentally ill people can say and do anything."*

D) Visibility.

Approximately 55% of the respondents agreed that they could identify someone with mental illness by the way s/he acts or talks. However, there was a mixed response to whether they could or could not recognize the symptoms of mental illness. During in-depth interviews, the attendants mentioned that a common misperception is that anyone with a mental

illness is "mad". They report that people with a mental illness usually talk very irrationally, do not take care of their appearance, and can often become aggressive. According to the attendants,

> "They behave poorly, talk abusively, and break things. He even beat me sometimes. This is controllable now, but sometimes behaves like an insane person."

One attendant mentioned that,

> "People go to traditional healer for mental illness because they think mentally ill people are an outcome of evil company"

### Reverse-scored domains

The mean scores and the standard deviation of the reverse-scored domains found were: treatability (5.2), recovery (5.3), and professional efficiency (6.4), as depicted in Fig 2.

A) Treatability.

More than 68% of the respondents agreed that there were medications for mental illness that would allow those who suffered from it to return to normal and productive lives, as shown in Table 3.

Data from in-depth interviews with attendants suggested that they believed that mental illness is treatable, and most of them were very satisfied with the services provided by the doctors and nurses. However, they all realized that there was a shortage of trained and skilled staff at the hospital, as well as poor management in the facilities. One respondent mentioned that,

> "Modern treatment was competent but time-consuming, which is why some relied solely on modern treatment at health facilities, while others also depended on traditional faith healers, as they did not have full faith in the medicines."

Though respondents agreed that medications were the cure for mental illness patients but health facilities do not have enough medication supply. According to the respondents,

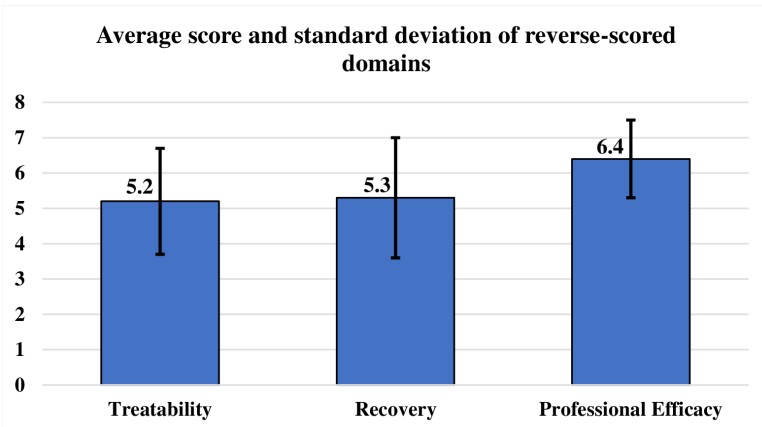

**Fig 2. Average score and standard deviation of reverse-scored stigma domains among the attendants of inpatients and outpatients, as per Day's Mental Illness Stigma Scale (n = 176).**

Table 3. Distribution of scores according to each of the reverse-scored domains.

| Domain | Statements | Agreement level | | | | | | |
|---|---|---|---|---|---|---|---|---|
| | | Complete disagree n (%) | Disagree n (%) | Partial disagree n (%) | Neutral n (%) | Partial agree n (%) | Agree n (%) | Complete agree n (%) |
| Treatability | Effective medications for mental illnesses helps to return to normal and productive lives | 1 (0.6) | 5 (2.8) | 18 (10.2) | 3 (1.7) | 28 (15.9) | 67 (38.1) | 54 (30.7) |
| | There are no effective treatments for mental illnesses | 66 (37.5) | 33 (18.8) | 20 (11.4) | 2 (1.1) | 19 (10.8) | 23 (13.1) | 13 (7.4) |
| | Little can be done to control the symptoms of mental illness | 70 (39.8) | 19 (10.8) | 18 (10.2) | 0 (0.0) | 34 (19.3) | 20 (11.4) | 15 (8.5) |
| Recovery | People with mental illness can never fully recover | 63 (35.8) | 43 (24.4) | 13 (7.4) | 5 (2.8) | 17 (9.7) | 25 (14.2) | 10 (5.7) |
| | People with mental illnesses remain ill for lifetime | 74 (42.0) | 43 (24.4) | 18 (10.2) | 5 (2.8) | 10 (5.7) | 18 (10.2) | 8 (4.5) |
| Professional efficiency | Psychiatrists and psychologists have the knowledge and skills needed to effectively treat [mental illnesses] | 2 (1.1) | 3 (1.7) | 10 (5.7) | 0 (0.0) | 10 (5.7) | 16 (9.1) | 135 (76.7) |
| | Professionals can provide effective treatments [of mental illnesses] | 4 (2.3) | 0 (0) | 8 (4.5) | 1 (0.6) | 13 (7.4) | 24 (13.6) | 126 (71.6) |

> "Doctors prescribe medicine but they do not get any medicine from the hospital as facilities had shortage of medicine supply. They have to buy medicine from the outside. He does not know about the supply of the medicine. All the medicines are available in the nearby pharmacies of the hospital."

B) Recovery

More than 35% of the respondents completely disagreed that people with a mental illness will remain ill for the rest of their lives and can never fully recover from it (Table 3).

In-depth exploration suggests that there is no effective treatment for mental illness. However, doctors, nurses, and one attendant mentioned that.

> "There are several patients who do get cured, but they have to continue visiting the doctors for regular check-ups, since there is always a chance of recurrence of the illness."

> "Mental illness is like a curse, a mysterious disease. It can happen to anyone without any reason, and can get cured without any reason also"

C) Professional efficiency.

More than 70% of the respondents completely agreed that psychologists and psychiatrists are knowledgeable and skilled enough to effectively treat mental illness and that they can provide effective treatment (Table 3).

During in-depth interviews, the participants also mentioned that they strongly believed that doctors had the necessary skills and expertise to treat mental illness. They thought that the nurses were also very capable of handling and communicating with the patients.

> "The number of service providers was not adequate for the hospital, since the doctors could not give enough time to the patients."

Exploration from the qualitative interview revealed that

> "Doctors come to all the patients every day and talk to them for enough time. They give advises to the both, patients and their attendants."

## Distribution of stigma scores according to sociodemographic characteristics

In Table 4, the results describe the distribution of stigma scores (from the Day's stigma scale) according to the socio-demographic characteristics. Relevant statistical analysis showed that there was no significant difference between the scores across the sociodemographic characteristics. However, with the increase in age, there was a decrease in stigma in all domains. It was also seen that males and those who were not educated were more stigmatized towards patients with a mental illness than females (Table 4). Distribution of the scores according to each domain is shown in Tables 2 and 3. According to the responses from the attendants of patients, it was found that there was a higher level of stigma and a negative attitude among the respondents towards the positively scored domains, and a lower level of stigma and a more positive attitude among the respondents towards the reverse-scored domains.

Further analysis showed that there is no statistically significant effect of the positive scored domain or statement on the agreement level of patients. This means the variation in the data is not explained by the domain. In the negative scored domain analysis revealed that the variability in the data is due to random variation or other external factors, and is not explained by the domain in the S1 Table.

The result showed in the Table 5 that, stigma around the positive scored domain like hygiene influenced by age (UB = 2.03, SB = 0.19, $P$ = 0.03) and anxiety influenced by the occupation (UB = 1.37, SB = 0.19, $P$ = 0.03) and both the results were statically significant. It also found that stigma in all domain is not statistically significant with the socio-demographic variables (Sex, education etc.). (Table 5)

## Discussion

This is the first study of stigma towards mental disorders among attendants of patients with mental illness in selected health facilities in Bangladesh. We found that the perception of attendants, who brought patients with mental illness to the health facilities, has stigma at both personal and perceived levels. The society treated the patient very differently and deleteriously. Results from our quantitative and qualitative data analysis were consistent and strongly support the above findings. For example, in terms of relationship segment it was found that most of the respondents agreed that it is difficult to maintain a normal relationship with someone who has mental illness which also supported by qualitative findings such as attendants of the patients mentioned that they treat the patients with compassion but, sometimes it feels like a burden or a responsibility.

**Table 4. Distribution of scores according to sociodemographic characteristics of the respondents from quantitative data collection.**

| Variable | Category | Positively scored domains [mean (SD)] | | | | Reverse-scored domains [mean (SD)] | | |
|---|---|---|---|---|---|---|---|---|
| | | Relationship | Hygiene | Anxiety | Visibility | Treatability | Recovery | Professional expertise |
| Age | ≤ 30 | 4.60 (1.91) | 5.10 (1.77) | 4.90 (1.39) | 5.34 (2.11) | 5.27 (1.72) | 5.44 (1.57) | 6.57 (0.88) |
| | 31-50 | 4.92 (1.35) | 5.59 (1.41) | 4.76 (1.07) | 5.13 (1.95) | 5.03 (1.34) | 5.08 (1.82) | 6.24 (1.22) |
| | ≥ 51 | 4.05 (1.77) | 5.98 (1.04) | 3.96 (0.82) | 4.03 (2.13) | 5.85 (1.25) | 5.95 (1.13) | 6.91 (0.20) |
| Sex | Male | 4.79 (1.56) | 5.47 (1.49) | 4.84 (1.21) | 5.39 (1.98) | 5.04 (1.54) | 5.09 (1.67) | 6.32 (1.17) |
| | Female | 4.69 (1.70) | 5.37 (1.63) | 4.66 (1.19) | 4.84 (2.10) | 5.33 (1.44) | 5.49 (1.73) | 6.52 (0.96) |
| Education | No schooling | 5.08 (1.15) | 5.83 (1.12) | 4.99 (0.96) | 5.30 (1.68) | 4.77 (1.36) | 4.88 (2.04) | 6.35 (0.88) |
| | Primary | 4.74 (1.72) | 5.69 (1.51) | 4.70 (1/09) | 5.16 (2.31) | 5.27 (1.50) | 5.34 (1.66) | 6.49 (1.07) |
| | Secondary and above | 4.69 (1.66) | 5.28 (1.61) | 4.74 (1.27) | 5.11 (2.02) | 5.21 (1.52) | 5.32 (1.66) | 6.39 (1.12) |
| Occupation | Farmer | 4.98 (1.20) | 5.96 (0.98) | 4.54 (0.85) | 6.41 (1.71) | 4.69 (1.16) | 4.93 (1.53) | 6.46 (0.77) |
| | Self employed | 4.81 (1.52) | 5.51 (1.57) | 4.87 (1.23) | 5.16 (2.05) | 4.94 (1.66) | 5.11 (1.86) | 6.36 (1.10) |
| | Service holder | 4.21 (1.95) | 4.92 (1.79) | 4.47 (1.33) | 5.17 (2.12) | 5.40 (1.73) | 5.63 (1.39) | 6.09 (1.43) |
| | Housewife | 4.89 (1.59) | 5.54 (1.53) | 4.79 (1.11) | 4.97 (2.03) | 5.27 (1.42) | 5.41 (1.70) | 6.53 (0.94) |
| | Unemployed | 4.75 (1.72) | 5.34 (1.37) | 4.88 (1.60) | 4.88 (1.97) | 5.38 (1.20) | 5.22 (1.63) | 6.56 (1.09) |
| | Others | 4.50 (1.64) | 4.56 (1.88) | 5.22 (0.71) | 4.34 (2.21) | 5.42 (1.37) | 3.63 (2.50) | 6.38 (1.25) |

**Table 5. Multivariate regression analyses between stigma score and age, sex, education, and occupation.**

| Domain | Unstandardized Coefficients | | Standardized Coefficients | t | p |
|---|---|---|---|---|---|
| | B | Std. Error | Beta | | |
| Age | 0.65 | 1.4 | 0.04 | .0.45 | 0.65 |
| Sex | -2.42 | 1.91 | -0.12 | -1.27 | 0.21 |
| Education | -1.47 | 1.69 | -0.032 | -0.87 | 0.39 |
| Occupation | 1.2 | 0.78 | 0.15 | 1.51 | 0.13 |
| **Hygiene** | | | | | |
| Age | 2.03 | 0.89 | 0.19 | 2.27 | 0.03* |
| Sex | -1.83 | 1.20 | -0.15 | -1.52 | 0.13 |
| Education | -1.39 | 1.06 | -0.11 | -1.31 | 0.19 |
| Occupation | 0.56 | 0.49 | 0.11 | 1.14 | 0.26 |
| **Anxiety** | | | | | |
| Age | -1.60 | 1.21 | -0.11 | -1.32 | 0.19 |
| Sex | -2.83 | 1.63 | -0.17 | -0.17 | 0.08 |
| Education | -1.97 | 1.44 | -0.11 | -1.37 | 0.17 |
| Occupation | 1.37 | 0.66 | 0.19 | 2.08 | 0.03* |
| **Visibility** | | | | | |
| Age | -1.08 | 0.89 | -0.10 | -1.21 | 0.23 |
| Sex | -1.86 | 1.19 | -0.15 | -1.55 | 0.12 |
| Education | -1.06 | 1.05 | -0.08 | -1.01 | 0.32 |
| Occupation | 0.32 | 0.48 | 0.06 | 0.66 | 0.51 |
| **Treatability** | | | | | |
| Age | 0.25 | 0.66 | 0.03 | 0.38 | 0.71 |
| Sex | 0.15 | 0.88 | 0.02 | 0.17 | 0.86 |
| Education | 0.25 | 0.78 | 0.03 | 0.32 | 0.75 |
| Occupation | 0.49 | 0.36 | 0.14 | 1.39 | 0.16 |
| **Recovery** | | | | | |
| Age | -0.23 | 0.50 | -0.04 | -0.45 | 0.64 |
| Sex | 0.88 | 0.67 | 0.13 | 1.31 | 0.19 |
| Education | 0.25 | 0.59 | 0.04 | 0.43 | 0.67 |
| Occupation | 0.02 | 0.27 | 0.01 | 0.05 | 0.96 |
| **Professional efficiency** | | | | | |
| Age | -0.42 | 0.32 | -0.11 | -1.32 | 0.19 |
| Sex | 0.72 | 0.42 | 0.17 | 1.70 | 0.09 |
| Education | 0.12 | 0.37 | -0.03 | -0.33 | 0.74 |
| Occupation | -0.16 | 0.17 | -0.09 | -0.95 | 0.34 |

Recent studies have shown that people have started to better understand and recognize mental illness. A study conducted among Bangladeshi university students showed that the students were very much aware of mental illness, though they had limited knowledge about mental illness [21]. However, in a narrative review of Ahad et. al. described that many negative impacts on people and communities are caused by the stigma associated with mental health issues and psychiatry, such as decreased treatment adherence, social isolation and discrimination, and delayed treatment-seeking behavior [22]. It is still difficult to change the thought process or attitude of the community, and even that of the attendants and family members of a person with mental illness [15]. The findings from this study also suggested that apart from the perceived stigma regarding the attitude of others towards patients with mental illness, their caregivers also had some personal

stigma concerning relationships, hygiene, visibility, and anxiety. A study conducted in Ethiopia that evaluated stigma experienced by patients with mental illness and their primary caregivers showed that perceived stigma is highly prevalent [23]. These stigmatized perceptions among the community are likely influenced by several factors such as stereotyped behavior, prejudice, and discrimination, which are observed more in developing countries [24,25]. The prevalence of perceived stigma among caregivers of people with severe mental illness was found to be higher than 89% in Ethiopia and similarly high in Morocco [26]. While administering a questionnaire adopted from the World Health Organization Family Interview Schedule stigma items, it was found that about 39% of the caregivers were worried about the visibility of patients. About 36% of the caregivers thought that they needed to hide the illness of the patients, while a similar number of them also expressed that they felt embarrassed and ashamed because of the patient in their family [27]. A qualitative study in South Africa identified that knowledge about mental health and the perceived stigma of care providers negatively influence the care that they are expected to deliver to patients with mental disorders [28].

At a personal level, this study showed that stigma was present in many aspects, with the highest stigma score of 5.4 towards hygiene issues of patients with mental illness. A study conducted in Bangladesh using the same stigma scale found that the second highest stigma score was in the relationship domain, which was 5.5, which is not consistent with our study findings [18]. A team of Egyptian researchers found that almost all caregivers of patients with mental illness suffered from moderate to severe burden as per the Zarit burden scale [29]. In China, the mean scores for stigma experiences according to the Modified Consumer Experiences of Stigma Questionnaire scale were 3 out of 5, implying that they faced general stigma [30]. In a study conducted in India, about 60% of the caregivers expressed a stigmatized attitude towards mental illness, which is similar to our study findings [31]. A study in Nepal also found similar results, where a medium level of stigma was found among the caregivers of patients with mental illness, who felt like they were looked down on by the community [32]. Interestingly, a study in Bangladesh found that 66% of the patients with mental illness thought that they received "good support from family," and they thought that there was a presence of stigma among their surrounding people after knowing about their mental illness [33].

In this study, most of the attendants mentioned that they had faith in the professional expertise of the service providers at the mental health facilities. Amanpreet Kaur et. al narrated in their study of mental health stigma that people have more faith and believe in traditional healers in terms of seeking care for mental health [34], which is dissimilar to our study findings. Furthermore, this study also showed that older age groups, females, and educated attendants were generally less stigmatized towards patients with mental illness, although there was no significant difference. A study in Ethiopia found that 98% of the caregivers perceived that medical interventions can cure mental illness, and that there was a statistically significant difference across the place of residency of the respondents [27]. Additionally, another study also showed that perceived stigma of caregivers and relatives of patients with mental illness was significantly associated with gender, marital status, place of residency, social support, and duration of relationship with the patient [26]. A national study in Lebanon showed that based on the Community Attitudes towards Mental Illness scores, the presence of stigma was significantly lower in females, university-level students, and those who were 18–24 years old, than in males, primary-level students, and those more than 70 years old [35]. However, a population-based study in China showed that females, older age groups, less educated, and retired or homemakers had higher public stigma [36]. Such variation in results may be due to differences in mental health awareness and literacy among people from different sociocultural settings. However, it implies that further studies need to be conducted, both at the community level and on specific populations (such as attendants of patients with mental illness), to understand the association of sociodemographic factors with perception and stigma towards mental illness and of patients with mental illness.

The findings suggest that, building on the understanding of treatability and expertise of mental health professionals in Bangladesh, the stigma and negative perceptions around relationships, hygiene, and other issues may be minimized through culture-sensitive health education messages reaching all tiers of the population.

## Strength and limitation of the study

The limitation of the study was that no specific mental illness and urban vs rural setting was selected. The sample was taken from health facilities where the respondents (attendants) brought patients with mental illness seeking treatment. As such, their stigmatized perceptions cannot be generalized to all attendants and caregivers of other patients with mental illness. The sampling technique for this study was convenient sampling and was not stratified according to age, gender, social status, and religion, which also makes it less generalizable. However, this study contains both qualitative and quantitative findings, which complement each other.

## Conclusion

The findings from this study suggested that the highest level of stigma among the attendants was associated towards the patients' ability to maintain relationships and hygiene. The respondents also expressed a negative attitude associated with the inability of the patients to maintain relationships and discretion regarding their illness and to humiliate the person with a mental illness, as well as their family. However, the attendants had a much more positive attitude towards the treatability, curability, and recovery of the patients due to their faith in the professional expertise of the service providers at the mental health facilities. It is recommended that studies at the community level, as well as on specific populations (such as attendants of patients with mental illness) are needed to be conducted. Further in-depth qualitative exploration to understand the effect of various factors on the perceived and personal stigma may help in developing interventions and counselling frameworks in the future.

## Supporting information

**S1 File. Original quantitative data set of this study.**
(SAV)

**S2 File. Original qualitative data set of this study.**
(XLSX)

**S1 Table. Multivariate analysis of this study.**
(PDF)

## Acknowledgments

Genuine gratitude is due to the Non-Communicable Disease Control (NCDC) Programme of the Directorate General of Health Services (DGHS) for this study's financial and technical support.

## Author contributions

**Conceptualization:** Salim M Chowdhury, AKM Fazlur Rahman, Saidur Rahman Mashreky.

**Data curation:** Shagoofa Rakhshanda, Labida Islam, Aklima Anwar Mitu, Abrar Wahab.

**Formal analysis:** Shagoofa Rakhshanda, Labida Islam, Aklima Anwar Mitu, Minhazul Abedin, Cinderella Akbar Mayaboti, Evan Atlantis.

**Funding acquisition:** Saidur Rahman Mashreky.

**Investigation:** AKM Fazlur Rahman, Evan Atlantis, Saidur Rahman Mashreky.

**Methodology:** Shagoofa Rakhshanda, Koustuv Dalal, Farah Naz Rahman, Salim M Chowdhury, AKM Fazlur Rahman.

**Project administration:** Labida Islam, Koustuv Dalal, Salim M Chowdhury, AKM Fazlur Rahman, Saidur Rahman Mashreky.

**Software:** Shagoofa Rakhshanda, Labida Islam.

**Supervision:** Saidur Rahman Mashreky.

**Writing – original draft:** Shagoofa Rakhshanda, Labida Islam, Aklima Anwar Mitu.

**Writing – review & editing:** Koustuv Dalal, Farah Naz Rahman, Minhazul Abedin, Abrar Wahab, Cinderella Akbar Mayaboti, Salim M Chowdhury, AKM Fazlur Rahman, Evan Atlantis, Saidur Rahman Mashreky.

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
