## [Decision Letter · Decision Letter 0]

23 May 2025

Dear Dr. Islam,

Thank you for submitting your manuscript to PLOS ONE. After careful consideration, we feel that it has merit but does not fully meet PLOS ONE’s publication criteria as it currently stands. Therefore, we invite you to submit a revised version of the manuscript that addresses the points raised during the review process.

We look forward to receiving your revised manuscript.

Kind regards,

Kshitij Karki, MPH, MA

Academic Editor

PLOS ONE

Journal Requirements:

The study was funded by the Directorate General of Health Services (DGHS) Bangladesh (invitation ref no: DGHS/LD/NCDC/Procurement plan/RPA (GOB) Service/2018-

19/ 2018/5217/SP-05) under the name of Saidur Rahman Mashreky as Principal  Investigator.

3. Please amend your list of authors on the manuscript to ensure that each author is linked to an affiliation. Authors’ affiliations should reflect the institution where the work was done (if authors moved subsequently, you can also list the new affiliation stating “current affiliation:….” as necessary).’

4. Please remove all personal information, ensure that the data shared are in accordance with participant consent, and re-upload a fully anonymized data set.

5. We are unable to open your Supporting Information file Data set.sav. Please kindly revise as necessary and re-upload.

Additional Editor Comments:

Please go through the suggestions from the reviewers and address. Thank you

Reviewers' comments:

Reviewer's Responses to Questions

**Comments to the Author**

1. Is the manuscript technically sound, and do the data support the conclusions?

Reviewer #1: Yes

Reviewer #2: Partly

2. Has the statistical analysis been performed appropriately and rigorously?

Reviewer #1: Yes

Reviewer #2: No

3. Have the authors made all data underlying the findings in their manuscript fully available?

Reviewer #1: Yes

Reviewer #2: No

4. Is the manuscript presented in an intelligible fashion and written in standard English?

Reviewer #1: Yes

Reviewer #2: No

Reviewer #1: This manuscript examines the stigma of attendants who bring mentally ill patients to a mental health facility in Bangladesh. Stigma was highest for hygiene issues (score: 5.4) and lowest for expertise (score: 6.4). There was more stigma for "relationships", "hygiene" and "visibility".

Overall it is well written. Some comments are listed below.

1. What are the mental illnesses of the patients in this study? Please also provide the characteristics of the subject patients.

2. Please provide details about the relationship between the attendant and the patient. (e.g., depth of relationship, duration of association, etc.)

Reviewer #2: The manuscript, “Personal and perceived stigma towards mental disorders among attendants of patients with mental illness in selected health facilities of Bangladesh,” addresses an important and underexplored aspect of mental health in low-resource settings. The topic is of considerable public health relevance, particularly in low- and middle-income countries like Bangladesh, where access to mental health care remains limited and stigma poses a major barrier to treatment. The mixed-methods design is appropriate, capturing both quantitative (n=176) and qualitative (n=20) perspectives. The use of the Day’s Mental Illness Stigma Scale, coupled with in-depth interviews, demonstrates an effort to understand nuanced perceptions across seven domains. Notably, the study highlights high levels of stigma in domains such as hygiene, relationships, and anxiety, while more positive attitudes were observed regarding treatability and trust in professional care—findings that could inform culturally sensitive interventions and awareness efforts.

Despite its strengths, the manuscript has several methodological and analytical limitations that should be addressed before it can be considered for publication. The use of convenience sampling, without adequate justification, introduces significant selection bias and limits the study’s generalizability. Additionally, the absence of stratification by key sociodemographic factors (e.g., urban/rural location, socioeconomic status, or diagnostic category) weakens the conceptualization of the study setting and undermines the representativeness of the findings. The rationale for the sample sizes in both components is not provided, and while the authors mention achieving thematic saturation in the qualitative data, no supporting documentation (e.g., saturation matrix or coding framework) is presented. Moreover, it is unclear whether the scale was validated for the local context. Providing details on translation, cultural adaptation, and psychometric validation—or citing appropriate sources—would strengthen the manuscript’s measurement validity.

The data analysis also requires enhancement. The manuscript presents only descriptive statistics, with no mention of inferential testing (e.g., p-values, confidence intervals) or specification of the statistical tests used to compare subgroups. The absence of multivariate analysis limits understanding of potential confounders, especially regarding the influence of age, sex, and education on stigma scores.

On the qualitative side, while the thematic findings are informative, the analysis lacks detail. The coding strategy, analytical framework, and measures to ensure reliability (such as double coding or inter-coder agreement) are not described. This limits confidence in the qualitative conclusions, which appear anecdotal rather than systematically derived. Furthermore, integration between qualitative and quantitative results is limited and should be improved to enrich interpretation.

Several sections of the manuscript would benefit from editing for clarity, conciseness, and grammar. The discussion section often reiterates results without deeper engagement or comparison with existing literature. Some citations are outdated or drawn from non-peer-reviewed sources; incorporating recent, peer-reviewed research would improve the scientific grounding. Ethical considerations also require greater detail—particularly regarding informed consent procedures, participant anonymity, and emotional safeguards during interviews. While the limitations are acknowledged, the discussion does not adequately address the implications of convenience sampling, diagnostic ambiguity, or cross-sectional design. These factors constrain the study’s capacity to make causal inferences and should be discussed more explicitly. Overall, the paper talks about an important topic and provides helpful initial findings, but it needs major revisions—especially in explaining the study design, improving analysis, and being clearer in reporting—to meet the publication standards of PLOS ONE. I believe that with these significant changes and clearer reporting, this study could substantially add to the research on mental health stigma in low-resource settings.

**Do you want your identity to be public for this peer review?** For information about this choice, including consent withdrawal, please see our Privacy Policy

Reviewer #1: No

Reviewer #2: No

---

## [Author Response · Author response to Decision Letter 1]

6 Jul 2025

Academic Editor

1. Please ensure that your manuscript meets PLOS ONE's style requirements, including those for file naming: Thank you for your feedback. Necessary revision was made according to the PLOS ONE’s requirements.

The study was funded by the Directorate General of Health Services (DGHS) Bangladesh (invitation ref no: DGHS/LD/NCDC/Procurement plan/RPA (GOB) Service/2018-19/ 2018/5217/SP-05) under the name of Saidur Rahman Mashreky as Principal Investigator. Please state what role the funders took in the study. If the funders had no role, please state: "The funders had no role in study design, data collection and analysis, decision to publish, or preparation of the manuscript." If this statement is not correct you must amend it as needed. Please include this amended Role of Funder statement in your cover letter; we will change the online submission form on your behalf: Thank you once again. The funders’ role was stated in the cover letter according to your suggestion.

3. Please amend your list of authors on the manuscript to ensure that each author is linked to an affiliation. Authors’ affiliations should reflect the institution where the work was done (if authors moved subsequently, you can also list the new affiliation stating “current affiliation:….” as necessary): Thank you. Author list was amended where the work was done and according to their current affiliation in the line number 8 to 18.

4. Please remove all personal information, ensure that the data shared are in accordance with participant consent, and re-upload a fully anonymized data set: Thank you once again. After removing all personal information re-upload was done.

5. We are unable to open your Supporting Information file Data set.sav. Please kindly revise as necessary and re-upload: Thank you once again. Necessary revision was made and re-upload was done.

Reviewer 1

1. What are the mental illnesses of the patients in this study? Please also provide the characteristics of the subject patients: Thank you for your comment. We did not explore the types of mental illness and include this in the limitation section in the line number 357.

According to your suggestion characteristics of the subject patients was included in the table 1 in the line number 168 to 175.

2. Please provide details about the relationship between the attendant and the patient. (e.g., depth of relationship, duration of association, etc.): Thank you once again. Depth of relationship was stated in the table 1 in the line number 173 to 174.

Reviewer 2

-The manuscript, “Personal and perceived stigma towards mental disorders among attendants of patients with mental illness in selected health facilities of Bangladesh,” addresses an important and underexplored aspect of mental health in low-resource settings. The topic is of considerable public health relevance, particularly in low- and middle-income countries like Bangladesh, where access to mental health care remains limited and stigma poses a major barrier to treatment. The mixed-methods design is appropriate, capturing both quantitative (n=176) and qualitative (n=20) perspectives. The use of the Day’s Mental Illness Stigma Scale, coupled with in-depth interviews, demonstrates an effort to understand nuanced perceptions across seven domains. Notably, the study highlights high levels of stigma in domains such as hygiene, relationships, and anxiety, while more positive attitudes were observed regarding treatability and trust in professional care—findings that could inform culturally sensitive interventions and awareness efforts.

1. Despite its strengths, the manuscript has several methodological and analytical limitations that should be addressed before it can be considered for publication. The use of convenience sampling, without adequate justification, introduces significant selection bias and limits the study’s generalizability. Additionally, the absence of stratification by key sociodemographic factors (e.g., urban/rural location, socioeconomic status, or diagnostic category) weakens the conceptualization of the study setting and undermines the representativeness of the findings. The rationale for the sample sizes in both components is not provided, and while the authors mention achieving thematic saturation in the qualitative data, no supporting documentation (e.g., saturation matrix or coding framework) is presented. Moreover, it is unclear whether the scale was validated for the local context. Providing details on translation, cultural adaptation, and psychometric validation—or citing appropriate sources—would strengthen the manuscript’s measurement validity:

• Thank you for your valuable comment. We selected ten tertiary level health facilities of Bangladesh among them two of the health facilities were specialized mental health care centres. These ten health facilities were all tertiary-level mental health centres in Bangladesh, where patients from across the country receive care. However, data from community settings may differ from facility data. Thus, we can say it is a national representative study; however, the study was less generalizable as data from community settings and in health facility settings are differ from each other. Thus, we include this in our study limitation settings in the line number 356 to 363.

• The justification for convenient sampling given in the method section in the line number 92 to 96 and in the limitation section in the line number 356 to 363.

• Back in 2019, as people are not that much aware about mental illness and we collected data from 2nd March 2019 to 20th April 2019, so during this time frame whatever patient we found with mental illness we included in this study considering their willing participation and the revision was made in the line number 87 to 88 and in the line number 99 to 100

• Qualitative data file was provided as a supplementary document in the line number 509.

• The scale was validated for the local context and necessary revision was made in the data collection instrument section in the line number 103 to 106.

2. The data analysis also requires enhancement. The manuscript presents only descriptive statistics, with no mention of inferential testing (e.g., p-values, confidence intervals) or specification of the statistical tests used to compare subgroups. The absence of multivariate analysis limits understanding of potential confounders, especially regarding the influence of age, sex, and education on stigma scores.

On the qualitative side, while the thematic findings are informative, the analysis lacks detail. The coding strategy, analytical framework, and measures to ensure reliability (such as double coding or inter-coder agreement) are not described. This limits confidence in the qualitative conclusions, which appear anecdotal rather than systematically derived. Furthermore, integration between qualitative and quantitative results is limited and should be improved to enrich interpretation:

• Thank you once again. Multivariate analysis was carried out as per your comment, we revised it in the line number 269 to 273 and include it in the supplementary document in the line number 510.

• Necessary changes were made in the qualitative data analysis section in the line number 147 to 152: “The in-depth interviews were transcribed verbatim and translated into English. The data was then summarized and analyzed manually by two researchers as per a guideline consisting of a list of a priori codes and themes, which were based on literature reviews. Any discrepancies or differences in opinions were discussed and resolved by the researchers. Thematic analysis was carried out in this study, which coincides with the quantitative results of this study. Relevant quotations were used to support the descriptive analysis of the data”.

3.Several sections of the manuscript would benefit from editing for clarity, conciseness, and grammar. The discussion section often reiterates results without deeper engagement or comparison with existing literature. Some citations are outdated or drawn from non-peer-reviewed sources; incorporating recent, peer-reviewed research would improve the scientific grounding. Ethical considerations also require greater detail—particularly regarding informed consent procedures, participant anonymity, and emotional safeguards during interviews. While the limitations are acknowledged, the discussion does not adequately address the implications of convenience sampling, diagnostic ambiguity, or cross-sectional design. These factors constrain the study’s capacity to make causal inferences and should be discussed more explicitly. Overall, the paper talks about an important topic and provides helpful initial findings, but it needs major revisions—especially in explaining the study design, improving analysis, and being clearer in reporting—to meet the publication standards of PLOS ONE. I believe that with these significant changes and clearer reporting, this study could substantially add to the research on mental health stigma in low-resource settings:

• Thank you again for your valuable comment. As per your instructions discussion section was revised in the line number 282 to 355.

• Citations were updated according to your comments and change was made in the reference section in the line number 379 to 505.

• According to your suggestion ethical clearance part was revised as follows “Written informed consent was obtained from each study participant. Participants had the right to withdraw from the study at any point of data collection. Attempts were made by the researchers to maintain scientific rigour, accuracy, impartiality, emotional safeguarding and anonymity where appropriate during data collection. Qualitative data was collected in a closed room to ensure the anonymity of the attendees” in the line number 155 to 160.

---

## [Decision Letter · Decision Letter 1]

20 Jul 2025

Dear Dr. Labida Islam,

We look forward to receiving your revised manuscript.

Kind regards,

Kshitij Karki, MPH, MA

Academic Editor

PLOS ONE

Journal Requirements:

Additional Editor Comments (if provided):

Please revise as per the suggestions of the reviewer. Thank you

Reviewers' comments:

Reviewer's Responses to Questions

**Comments to the Author**

Reviewer #1: All comments have been addressed

Reviewer #2: (No Response)

2. Is the manuscript technically sound, and do the data support the conclusions?

Reviewer #1: Yes

Reviewer #2: Partly

3. Has the statistical analysis been performed appropriately and rigorously?

Reviewer #1: Yes

Reviewer #2: No

4. Have the authors made all data underlying the findings in their manuscript fully available?

Reviewer #1: Yes

Reviewer #2: Yes

5. Is the manuscript presented in an intelligible fashion and written in standard English?

Reviewer #1: Yes

Reviewer #2: Yes

Reviewer #1: (No Response)

Reviewer #2: Thank you for the revised submission. The authors have addressed some earlier concerns and made improvements to the manuscript. The introduction now presents a clearer rationale for investigating stigma in the Bangladeshi context, and the overall language has improved. These revisions show a good-faith effort to engage with reviewer feedback and are appreciated.

However, several key issues remain unresolved. Some of the claims made in the author response, such as the inclusion of multivariate analysis and the validation of the stigma scale, are not supported by the actual content of the manuscript.

On lines 103 to 106, the authors mention that the Day’s Mental Illness Stigma Scale was translated into Bengali and back-translated into English by bilingual researchers. While this is an important step, the manuscript provides no evidence of psychometric validation in the Bangladeshi context. There is no mention of reliability statistics, such as Cronbach’s alpha, no description of how the scale was culturally adapted, and no citation of any prior validation study. Since the scale forms the core of the study’s quantitative component, this omission limits the reliability of the findings.

Reference 17, which the authors cite in support of the scale, appears to point to a general Google Scholar search rather than a valid source. If the intention was to cite the Bangladesh National Mental Health Survey, that reference needs to be correctly formatted and clearly explained. Even if the survey used the Day’s scale, its use alone does not imply validation. If validation was conducted, the authors should cite the appropriate report and describe the findings. Otherwise, they should consider conducting a basic psychometric analysis themselves or referencing another study, such as the one listed as reference 26 or similar publications that have used the scale in Bangladesh. The current citation does not provide adequate justification for the scale’s validity.

In their response, the authors state that multivariate analysis was performed. However, the revised manuscript does not include any such analysis. The results section presents only descriptive statistics, including means, standard deviations, and frequencies. No regression models, p-values, confidence intervals, or adjusted comparisons are provided. This is a major gap, especially since the authors draw interpretive conclusions across demographic groups.

An ANOVA is included in the supplementary file (S1 Table), but this only compares mean scores across stigma domains within the scale. It does not examine how respondent characteristics, such as education or gender, influence stigma scores. This was the focus of the requested multivariate analysis and remains unaddressed. Without this level of analysis, it is not possible to understand the independent contributions of key variables or to control for confounding.

The methods section notes that SPSS was used, but the actual statistical procedures are not described. For example, Table 3 presents stigma scores by education level, but the manuscript does not report whether the differences are statistically significant. Statements which suggests that participants with higher education showed less stigma, are not supported by formal statistical testing.

The qualitative component also lacks sufficient methodological detail. While the manuscript states that thematic saturation was reached after 20 interviews, no explanation is provided for how this decision was made. There is no mention of how saturation was defined or assessed, and there is no coding framework, saturation matrix, or supporting evidence. This makes it difficult to evaluate the rigor of the analysis.

The authors also do not describe how codes were developed, how many researchers were involved in the coding process, or whether inter-coder reliability was assessed. There is no mention of using any established qualitative analysis framework. Without these details, it is hard to judge the trustworthiness of the qualitative findings. Additionally, the integration of qualitative and quantitative results is minimal. The findings are presented separately rather than used to build a coherent narrative that draws on both data types.

The sampling approach also remains a concern. The study uses convenience sampling from three health facilities. While this is acknowledged briefly, its implications are not discussed in sufficient depth. There is no reflection on how this may affect representativeness or introduce bias. The authors also do not stratify their analysis by meaningful variables, such as type of mental illness, rural versus urban setting, or the relationship between the attendant and the patient. These factors could provide important context for interpreting stigma but are not explored.

Lastly, some language in the manuscript still implies causation, which is inappropriate for a cross-sectional study. Since the data are observational, the conclusions should be limited to associations.

In summary, while the study addresses an important topic and the revisions show progress, significant methodological concerns remain. The absence of multivariate analysis, and the limited detail in both statistical and qualitative methods reduce the strength and clarity of the findings. I encourage the authors to address these issues more thoroughly in a future revision.

My recommendation is major revision.

**Do you want your identity to be public for this peer review?** For information about this choice, including consent withdrawal, please see our Privacy Policy

Reviewer #1: **Yes: ** Tetsuji Kitano

Reviewer #2: No

---

## [Author Response · Author response to Decision Letter 2]

8 Sep 2025

Thank you for the revised submission. The authors have addressed some earlier concerns and made improvements to the manuscript. The introduction now presents a clearer rationale for investigating stigma in the Bangladeshi context, and the overall language has improved. These revisions show a good-faith effort to engage with reviewer feedback and are appreciated.

However, several key issues remain unresolved. Some of the claims made in the author response, such as the inclusion of multivariate analysis and the validation of the stigma scale, are not supported by the actual content of the manuscript.

1. On lines 103 to 106, the authors mention that the Day’s Mental Illness Stigma Scale was translated into Bengali and back-translated into English by bilingual researchers. While this is an important step, the manuscript provides no evidence of psychometric validation in the Bangladeshi context. There is no mention of reliability statistics, such as Cronbach’s alpha, no description of how the scale was culturally adapted, and no citation of any prior validation study. Since the scale forms the core of the study’s quantitative component, this omission limits the reliability of the findings.

-Thank you for your valuable comment. According to your suggestion necessary revision was made in the line number 106 to 109. Prior to data collection, a pretest of the questionnaire was conducted in the above-selected health facilities among 30 attendants. The scale was utilized in the study since there aren't any culturally validated measures that measure the stigma associated with mental illness in Bangladesh.

Study conducted in Bangladesh using the same scale showed that prior to the use in their study, the scale was administered to a sample of 50 participants (excluding the main sample) in both rural and urban areas to understand the comprehensibility of items. No item was found to be difficult to understand, therefore, no change was required. The Cronbach’s alpha for the total scale in the present study was 0.67. The Cronbach’s alphas for the subscales in the present study were 0.71, 0.50, 0.52, 0.76, 0.46, 0.52, and 0.74, respectively. They assessed test–retest reliability of the scale on a sample of an additional 100 participants in both settings with a gap of 2 weeks. The test–retest reliability coefficient for the present study was found to be r = 0.81 at p < 0.01.

2. Reference 17, which the authors cite in support of the scale, appears to point to a general Google Scholar search rather than a valid source. If the intention was to cite the Bangladesh National Mental Health Survey, that reference needs to be correctly formatted and clearly explained. Even if the survey used the Day’s scale, its use alone does not imply validation. If validation was conducted, the authors should cite the appropriate report and describe the findings. Otherwise, they should consider conducting a basic psychometric analysis themselves or referencing another study, such as the one listed as reference 26 or similar publications that have used the scale in Bangladesh. The current citation does not provide adequate justification for the scale’s validity.

-Thank you once again. Reference 17 was corrected and revised and another reference was put with 17 no reference which was previously 26 number but now it was 18 number in the line number 106 to 107.

The scale was used in National Mental Health survey but they only validated the tool by translation and back-translation. But in another study, they used Cronbach’s alpha test. prior to the use in their study, the scale was administered to a sample of 50 participants (excluding the main sample) in both rural and urban areas to understand the comprehensibility of items. No item was found to be difficult to understand, therefore, no change was required. The Cronbach’s alpha for the total scale in the present study was 0.67. The Cronbach’s alphas for the subscales in the present study were 0.71, 0.50, 0.52, 0.76, 0.46, 0.52, and 0.74, respectively. They assessed test–retest reliability of the scale on a sample of an additional 100 participants in both settings with a gap of 2 weeks. The test–retest reliability coefficient for the present study was found to be r = 0.81 at p < 0.01.

3. In their response, the authors state that multivariate analysis was performed. However, the revised manuscript does not include any such analysis. The results section presents only descriptive statistics, including means, standard deviations, and frequencies. No regression models, p-values, confidence intervals, or adjusted comparisons are provided. This is a major gap, especially since the authors draw interpretive conclusions across demographic groups.

An ANOVA is included in the supplementary file (S1 Table), but this only compares mean scores across stigma domains within the scale. It does not examine how respondent characteristics, such as education or gender, influence stigma scores. This was the focus of the requested multivariate analysis and remains unaddressed. Without this level of analysis, it is not possible to understand the independent contributions of key variables or to control for confounding.

-Thank you for your valuable comment. As per your suggestion multivariate analysis was carried out and we put it as table 5 in the line number 304 to 310.

4. The methods section notes that SPSS was used, but the actual statistical procedures are not described. For example, Table 3 presents stigma scores by education level, but the manuscript does not report whether the differences are statistically significant. Statements which suggests that participants with higher education showed less stigma, are not supported by formal statistical testing.

-Thank you. Necessary revision was made in the line number 304 to 310.

5. The qualitative component also lacks sufficient methodological detail. While the manuscript states that thematic saturation was reached after 20 interviews, no explanation is provided for how this decision was made. There is no mention of how saturation was defined or assessed, and there is no coding framework, saturation matrix, or supporting evidence. This makes it difficult to evaluate the rigor of the analysis.

The authors also do not describe how codes were developed, how many researchers were involved in the coding process, or whether inter-coder reliability was assessed. There is no mention of using any established qualitative analysis framework. Without these details, it is hard to judge the trustworthiness of the qualitative findings. Additionally, the integration of qualitative and quantitative results is minimal. The findings are presented separately rather than used to build a coherent narrative that draws on both data types.

• Thank you once again. As qualitative data collecting was no longer generating new themes, insights or any codes relevant to our research question and the redundancy was consistent, with the judgement of lead researcher the qualitative study team decided to stop data collection after 40 interviews. This thematic saturation aligns with recommendations from Guest et al. that data saturation can often be achieved within 12-15 interviews for relative similar participant groups. The necessary revision was made in the line number 120 to 124. Supporting evidence was given in the supplementary document S2 File in the line number 540.

• To enhance reliability, two researchers independently coded four of the same initial transcripts, and the results were compared for the establishment of the codes in the line number 159 to 161.

• Codes were developed starting with a broad and descriptive familiarization. Then they were refined into meaningful categories and then grouped into themes in the line number 157 to 159. After that, a documentation was made with a codebook, which has been provided in the supplementary document.

• Inter-coder agreement was assessed periodically by the joint coding an additional transcript every 5 interviews. (Cohen’s kappa calculation to monitor consistency where k= 0.82 which indicates substantial agreement in the line number 162 to 163.

• Qualitative data analysis followed Braun and Clarke six phase framework for thematic analysis and revision was made in the line number 162 to 163.

• Integration was done according to your suggestion in the line number 227 to 229, 240, 258 to 259 and 279.

6. The sampling approach also remains a concern. The study uses convenience sampling from three health facilities. While this is acknowledged briefly, its implications are not discussed in sufficient depth. There is no reflection on how this may affect representativeness or introduce bias. The authors also do not stratify their analysis by meaningful variables, such as type of mental illness, rural versus urban setting, or the relationship between the attendant and the patient. These factors could provide important context for interpreting stigma but are not explored.

-Thank you for your comment. We selected ten tertiary level health facilities of Bangladesh among them two of the health facilities were specialized mental health care centres, while the rest of eight medical college hospitals were selected based on the availability of mental health services. These Ten health facilities were all tertiary-level mental health centres in Bangladesh, where patients from across the country receive care in the line number 88 to 92.

In Bangladesh these ten tertiary facilities are the only facilities who treat patients with mental illness and as we mentioned before the two facilities are the only specialized mental health institutes. In this study we included all of them thus we can say it a representative sample.

The relationship between the attendant and the patient was discussed in the table 1 in the line number 187 to 188. Though we only explore this relationship for the qualitative part.

Type of mental illness and urban vs rural was not explored in this study and we include this limitation in the limitation part in the line number 386 to 388.

7. Lastly, some language in the manuscript still implies causation, which is inappropriate for a cross-sectional study. Since the data are observational, the conclusions should be limited to associations.

-Thank you once again for your valuable comment. Revision was made according to your suggestion in the line number 395 to 399.

---

## [Decision Letter · Decision Letter 2]

21 Sep 2025

Personal and perceived stigma towards mental disorders among attendants of patients with mental illness in selected health facilities of Bangladesh

PONE-D-24-40944R2

Dear Dr. Labida Islam,

We’re pleased to inform you that your manuscript has been judged scientifically suitable for publication and will be formally accepted for publication once it meets all outstanding technical requirements.

Kind regards,

Kshitij Karki, MPH, MA

Academic Editor

PLOS ONE

Additional Editor Comments (optional):

Reviewers' comments:

Reviewer's Responses to Questions

**Comments to the Author**

Reviewer #2: All comments have been addressed

2. Is the manuscript technically sound, and do the data support the conclusions?

Reviewer #2: Yes

3. Has the statistical analysis been performed appropriately and rigorously?

Reviewer #2: Yes

4. Have the authors made all data underlying the findings in their manuscript fully available?

Reviewer #2: Yes

5. Is the manuscript presented in an intelligible fashion and written in standard English?

Reviewer #2: Yes

Reviewer #2: I would like to express my appreciation to the authors for their careful and thoughtful revisions. The revised manuscript shows clear improvements in both methodological transparency and analytical rigor.

Key concerns raised in the prior review, including the lack of psychometric validation for the stigma scale, absence of multivariate analysis, and insufficient detail in the qualitative methodology, have now all been addressed comprehensively. The authors have included appropriate reliability measures, regression modeling, and a well-articulated qualitative coding framework, including inter-coder agreement and use of thematic analysis. Importantly, they have also improved the integration between quantitative and qualitative findings, clarified the representativeness of their sampling strategy, and revised the discussion to reflect the observational nature of the data.

These changes significantly strengthen the credibility and contribution of the study. The paper now presents a compelling mixed-methods exploration of stigma toward mental illness in the Bangladeshi context, a topic of high relevance but limited coverage in the current literature. The author's revisions reflect sincere engagement with reviewer feedback and enhance the value of the work for both academic and public health audiences. I am satisfied with the quality of the revised submission and recommend the manuscript for publication in PLOS ONE.

I wish the authors the best of luck in their future scientific endeavors.

**Do you want your identity to be public for this peer review?** For information about this choice, including consent withdrawal, please see our Privacy Policy

Reviewer #2: No

---

## [Editor Report · Acceptance letter]

PONE-D-24-40944R2

PLOS ONE

Dear Dr. Islam,

I'm pleased to inform you that your manuscript has been deemed suitable for publication in PLOS ONE. Congratulations! Your manuscript is now being handed over to our production team.

Kind regards,

on behalf of

Dr. Kshitij Karki

Academic Editor

PLOS ONE